# Associations of Mutually Exclusive Categories of Physical Activity and Sedentary Behavior with Body Composition and Fall Risk in Older Women: A Cross-Sectional Study

**DOI:** 10.3390/ijerph20043595

**Published:** 2023-02-17

**Authors:** Renoa Choudhury, Joon-Hyuk Park, Chitra Banarjee, Ladda Thiamwong, Rui Xie, Jeffrey R. Stout

**Affiliations:** 1Department of Mechanical Engineering, University of Central Florida, Orlando, FL 32816, USA; 2Disability, Aging and Technology Cluster, University of Central Florida, Orlando, FL 32816, USA; 3College of Medicine, University of Central Florida, Orlando, FL 32816, USA; 4College of Nursing, University of Central Florida, Orlando, FL 32816, USA; 5Department of Statistics and Data Science, University of Central Florida, Orlando, FL 32816, USA; 6School of Kinesiology and Physical Therapy, College of Health Professions and Sciences, University of Central Florida, Orlando, FL 32816, USA

**Keywords:** aging, accelerometry, physical activity, sedentary behavior, bioelectrical impedance analysis, fall risk, older adults

## Abstract

The individual effects of physical activity (PA) and sedentary behavior (SB) on health are well-recognized. However, little is known about the extent to which different combinations of these behaviors are associated with body composition and fall risk in older adults. This cross-sectional study examined the associations of mutually exclusive categories of PA and SB with body composition and fall risk in older women. Accelerometer-measured PA, body composition and fall risk (static and dynamic balance) parameters were assessed among 94 community-dwelling older women. The participants were categorized into four groups: active-low sedentary, active-high sedentary, inactive-low sedentary and inactive-high sedentary (active: ≥150 min/week moderate-to-vigorous PA (MVPA); low sedentary: lowest tertile of SB and light PA ratio). Compared to the inactive-high sedentary group, more favorable body composition and dynamic balance results were found in the active-low sedentary (body fat mass index (BFMI): *β* = −4.37, *p* = 0.002; skeletal muscle mass index (SMI): *β* = 1.23, *p* = 0.017; appendicular lean mass index (ALMI): *β* = 1.89, *p* = 0.003; appendicular fat mass index (AFMI): *β* = −2.19, *p* = 0.003; sit-to-stand: *β* = 4.52, *p* = 0.014) and inactive-low sedentary (BFMI: *β* = −3.14, *p* = 0.007; SMI: *β* = 1.05, *p* = 0.014; AFMI: *β* = −1.74, *p* = 0.005, sit-to-stand: *β* = 3.28, *p* = 0.034) groups. Our results suggest that PA programs focusing on concurrently achieving sufficient MVPA and reduced SB might promote a healthy body composition and reduced fall risk among older adults.

## 1. Introduction

Physical activity (PA) is vital for healthy aging and impeding many age-related reductions in health and physical functioning [1]. The World Health Organization (WHO) 2020 PA guidelines for older adults recommend at least 150 min of moderate-intensity aerobic activity or 75 min of vigorous-intensity aerobic activity, or an equivalent combination of both activities per week [2]. Studies have shown that engaging in sufficient moderate-to-vigorous PA (MVPA), i.e., ≥150 min/week of MVPA, contributes to the maintenance of a healthy weight and better physiological cardiovascular adaptations, muscle strength and physical functioning in older age [3,4,5]. In contrast, a growing body of evidence suggests that higher levels of sedentary behavior (SB) or proxy measures, such as television viewing, are associated with increased risks of obesity [6], metabolic syndrome [7], functional impairment [8] and all-cause mortality among older adults [9], regardless of meeting the PA guidelines. SB (i.e., any activities characterized by energy expenditure ≤1.5 metabolic equivalents while in a seated, reclined or lying posture [10]) is considered a distinct behavioral entity from physical inactivity (i.e., not meeting 150 min/week of MVPA [11]). Therefore, it could be possible for an individual to be physically active (i.e., meeting MVPA recommendations) but still spend prolonged periods in SB throughout the rest of the day. Conversely, some individuals may not be physically active yet may accumulate lower levels of sedentary time by engaging in light PA (LPA) throughout the day.

The prior studies have mostly focused on exploring the independent associations of SB, LPA and MVPA with body composition and fall risk among older adults, rather than investigating the daily equilibrium between these constructs [12,13,14,15,16,17,18]. However, the extent to which different combinations of these behaviors (i.e., active-low sedentary, active-high sedentary, inactive-low sedentary or inactive-high sedentary) may influence the body composition and fall risk remains poorly understood. It is to be noted that the medical costs related to older adult fall injuries pose a substantial financial burden to our healthcare system. In 2015, the estimated medical costs associated with both fatal and non-fatal falls reached approximately $50 billion in the USA [19]. Understanding how the body composition and fall risk in older adults are influenced by mutually exclusive categories of PA and SB would shed light into what are the best PA strategies to minimize age-related adverse changes in body composition and reduce the medical and rehabilitation expenditure associated with older adult falls.

To date, a small number of studies have utilized mutually exclusive PA categories to understand the combined effects of PA and SB on different health markers [20,21,22,23]. For instance, in population-based samples from the United States and England, it was observed that physically active adults, regardless of their sedentary status, had better cardiometabolic health profiles in comparison to those not meeting the MVPA guidelines [20,21]. Another study from the Osteoarthritis Initiative found that adults with knee osteoarthritis who did not meet the MVPA guidelines had a higher risk of developing functional limitations compared to those classified as physically active, irrespective of their sedentary status [22]. However, to our knowledge, no study has yet investigated the associations of such PA behavioral categories with body composition and fall risk in older adults.

The aim of this study is to examine the associations of four mutually exclusive categories of PA and SB with body composition and fall risk in a sample of community-dwelling older US women. In this study, the MVPA statuses were classified as ‘active’ and ‘inactive’ based on meeting thresholds of time spent in MVPA in bouts lasting ≥1 min [24]. A distribution-based approach was used to classify the sedentary status as ‘low’ or ‘high’, depending on the time spent in sedentary behavior and LPA [21]. Based on previous findings from the abovementioned studies [20,21,22], we hypothesized that physically active participants, regardless of their sedentary status, would exhibit a more favorable body composition profile and reduced fall risk than those categorized as being inactive.

## 2. Materials and Methods

### 2.1. Participants and Study Design

In this cross-sectional study, 94 community-dwelling older women (aged between 60 and 96 years) were recruited from the Central Florida region, USA, between February and August 2021. The details of this study design and sampling procedure have been described elsewhere [25]. The inclusion criteria were: (i) aged 60 years or above; (ii) able to walk (with or without assistive devices such as canes or crutches, but not requiring assistance from another person); (iii) no marked cognitive impairment (i.e., Memory Impairment Screen score ≥5 [26,27]); (iv) living in their own homes or apartments; (v) fluent in English or Spanish. The exclusion criteria were: (i) having a medical condition that may preclude participation in balance tests (e.g., inability to stand on the balance plate) and PA (e.g., shortness of breath, dizziness, tightness in the chest or unusual fatigue at light exertion); (ii) currently receiving treatment from a rehabilitation facility; (iii) having medical implants (e.g., pacemakers). The study was approved by the Institutional Review Board at the University of Central Florida (Protocol No: 2189; 10 September 2020). This study required two visits to the laboratory, separated by a 7-day PA monitoring period in free-living conditions. In the first visit, the participants completed the informed consent form and self-reported their age and race, followed by the fall risk assessment tests. At the end of the visit, each participant was fitted with an accelerometer and given instructions on how to wear it during the PA monitoring period. After 7 days, the second visit took place, in which participants returned the accelerometers and went through anthropometric and body composition measures.

### 2.2. Measurements

#### 2.2.1. Fall Risk Assessment

Fall risk was assessed using static and dynamic balance tests. Static balance was measured using a BTrackS Balance System (Balance Tracking Systems, San Diego, CA, USA), consisting of a portable BTrackS Balance Plate and BTrackS Assess Balance Software running on a computer. The BTrackS Balance Plate is an FDA-registered, lightweight (<7 kg) force plate for measuring center of pressure (COP) excursions during the static stance. It has demonstrated excellent validity using Pearson’s product moment correlations (r > 0.90) and high test–retest reliability (intraclass correlation coefficient, ICC = 0.83) [28]. The balance test was done using a standardized protocol, comprising four 20 s trials with minimal inter-trial delays (<10 s). During the test, the participants were asked to stand as still as possible on the force plate with their hands on their hips, eyes closed and feet shoulder-width apart. Each trial started and ended with an auditory tone. The first trial was performed for familiarization only and the remaining three trials were used to calculate the average COP path length (in cm) across trials using the BTrackS Assess Balance Software. The COP path length is a proxy measure for the postural sway magnitude, and larger values of the COP path length have been associated with greater postural sway, indicating poorer static balance [29].

Dynamic balance was assessed using the 30 s sit-to-stand test (also known as the ‘chair stand test’) [30]. The participants were instructed to fold their arms across their chest, stand away from a chair and return to a sitting position as many times as possible within 30 s. If a participant used their hand during the test, they scored zero.

In addition, to understand the perceived risk of falling among participants, the fear of falling was assessed using the Short Falls Efficacy Scale—International (FES-I) questionnaire. The short FES-I is a 7-item, easy-to-administer tool that measures the level of concern about falling while performing seven activities [31]. The level of concern is measured on a 4-point Likert scale (1 = not at all concerned to 4 = very concerned), and the total scores range from 7 to 28. Short FES-I scores of 7–10 indicate low concern of falling, while scores of 11–28 indicate high concern of falling.

#### 2.2.2. PA Assessment

The participants were instructed to wear the ActiGraph GT9X Link instrument (ActiGraph LLC., Pensacola, FL, USA) on their non-dominant wrists for seven consecutive days in a free-living environment. The GT9X Link contains a tri-axial MEMS accelerometer (i.e., the same validated accelerometer used in previous-generation ActiGraph models, including the ActiGraph wGT3X-BT [32]), with a dynamic range of ±8 gravitational units (g). The ActiGraph accelerometer has shown good reliability in assessing free-living physical activity (intraclass correlation coefficient, ICC = 0.97) [33] and has been validated against oxygen consumption in laboratory conditions (Pearson’s correlation coefficient, r  =  0.810, *p*  <  0.001) [34]. The participants were asked to wear it during all waking hours and remove it only during sleeping, showering, swimming and medical imaging tests. Once the 7-day PA monitoring period was completed, the raw data from the GT9X accelerometer were downloaded and converted to “.csv” files using ActiLife 6 software v6.13.4 (ActiGraph LLC, Pensacola, FL, USA).

The data processing was done in R statistical software (R Core Team, Vienna, Austria) using GGIR package (version 2.4-0) [35]. The data processing steps included: (i) the autocalibration of acceleration signals according to local gravity [36]; (ii) non-wear time detection; (iii) the calculation of the Euclidean norm (i.e., vector magnitude) of acceleration minus 1 g (ENMO), expressed in milli-gravitational units or mg (described in detail elsewhere [37]). Only days during which the accelerometer was worn for at least 14 h were counted as valid days of data. At least six valid days of data were required for a participant to be included in the analysis. The time periods spent in SB, LPA and MVPA were estimated using the following non-dominant wrist-specific ENMO cut-off points for older adults, adopted from the literature: (i) SB < 30 mg; (ii) 30 mg ≤ LPA < 100 mg; (iii) MVPA ≥ 100 mg [38,39].

#### 2.2.3. Body Composition Assessment

The body composition was assessed using a portable bioelectrical impedance analysis device (InBody S10, InBody Corp., Seoul, Korea), as per the manufacturer’s guidelines. The InBody technology has shown a high correlation with dual-energy X-ray absorptiometry (DXA), i.e., the gold standard for body composition measurements, for assessing the appendicular lean mass (Pearson’s correlation coefficient, r = 0.86), fat-free mass (r = 0.93) and percentage of body fat (r = 0.92) in older adults [40]. The participants were instructed to avoid exercise for 6–12 h, eating for 3–4 h and drinking alcohol and coffee for 24 h prior to testing. The body composition measurements were taken during late morning, and the participants were asked to empty their bladder and remove their socks, shoes and metal objects (e.g., watches, jewelry) before testing. Touch-type electrodes were placed on eight precise tactile points of the body (i.e., two electrodes on each ankle, one on each middle-finger and one on each thumb) in a seated position to achieve a multi-segmental frequency analysis. The time required to complete bioelectrical impedance measurements for each participant was around 2 min, and the body composition data were immediately analyzed. For each test, the whole-body and segmental (i.e., trunk, both arms and legs) muscle, fat and water values were obtained from the printed results sheet, which included intracellular water, extracellular water, total body water, dry lean mass, fat-free mass, fat mass and skeletal muscle mass results. The body mass was measured in kilograms with no shoes in a minimally clothed state using a digital scale, and the height was measured in cm using a stadiometer. The body mass index (BMI) was calculated as the weight (kg) divided by the square height (m^2^).

### 2.3. Mutually Exclusive Categories of PA and SB

The ratio of average sedentary time (min/day) to average LPA time (min/day) was used to classify sedentary statuses based on the existing literature [20,21,41]. Unlike MVPA guidelines, currently no recommended thresholds exist for SB and LPA; therefore, a data-driven approach was employed, and the participants were divided into tertiles based on their sedentary time-to-LPA time ratio. Given the high prevalence of sedentary lifestyles among older adults [42,43,44], the participants were classified as ‘low sedentary’ if they resided in the first tertile and ‘high sedentary’ if they resided in the remaining tertiles. The MVPA status was categorized as ‘active’ or ‘inactive’ based on meeting the weekly threshold 150 min of MVPA [24]. Then, the categories of SB and MVPA statuses were combined to formulate the following four mutually exclusive behavioral categories: (1) active-low sedentary, (2) active-high sedentary, (3) inactive-low sedentary and (4) inactive-high sedentary [20,21,22].

### 2.4. Statistical Analysis

All statistical analyses were performed in R statistical software (version 4.1.2, R Core Team, Vienna, Austria) with a statistical significance level (*p*) of 0.05. The participants’ characteristics are presented as means (standard deviation, SD) for continuous variables and percentages (frequency) for categorical variables, stratified by each behavioral category. The normality of the outcome variables (i.e., body composition and balance scores) was checked using the Shapiro–Wilk test. The results from the body composition and fall risk assessment were summarized as means (SD) for normally distributed variables and medians (Interquartile range, IQR) for non-normal distributions. Differences across groups were examined using a one-way analysis of variance (ANOVA) and Kruskal–Wallis test for normally and non-normally distributed data, respectively. To test our hypothesis, multiple linear regression models were fitted for each outcome variable using behavioral categories as the independent variables, and age, race and accelerometer wear times as covariates. The a priori sample size calculation revealed that the minimum number of samples for 7 explanatory variables at a statistical power level of 0.8 and a medium effect size (Cohen f^2^ = 0.2) would be 79; therefore, our sample size (i.e., *N* = 94) was sufficient for multiple regression. The least desirable behavioral category (i.e., inactive-high sedentary) was selected as the reference group in the regression analysis.

## 3. Results

Ninety-one participants were included in the analysis, after retaining only those who had at least six valid days of accelerometer data and completed the body composition assessment. The mean (SD) age of the participants was 74.9 (7.35) years and the majority of the participants were white (74%). The mean (SD) BMI was 26.8 (5.49) kg/m^2^ and the mean (SD) accelerometer wear period was 16.8 (2.03) hours/day among the sample. As shown in Figure 1, 19% of the participants were active and low sedentary (*n* = 17), 13% were active but had high sedentary status (*n* = 12), 32% were inactive but low sedentary (*n* = 29) and 36% were inactive and high sedentary (*n* = 33). When stratified by age groups, the participants aged between 60 and 69 years (*n* = 23) were widely distributed among four behavioral categories. In the ‘70–79 years’ age group (*n* = 47), the proportion of inactive participants was higher than their active counterparts (66% vs. 34%). Among the participants aged 80 years or above (*n* = 21), 86% did not meet the 150 min/week of MVPA recommendation (inactive-high sedentary: 48%; inactive-low sedentary: 38%). Table 1 presents the characteristics of the study participants according to mutually exclusive PA categories.

In Figure 2, the variations in PA patterns over 24 hours by mutually exclusive PA categories are presented, expressed as vector magnitude (VM) counts in 60 s epoch data, i.e., counts per minute (cpm). Overall, the activity levels across all groups were generally low during night hours, then showed a substantial increase during morning hours and gradually decreased more or less noticeably as the day progressed and evening approached. The active-high sedentary group reached their peak activity levels in the early morning (between 7.00 am and 8.00 am), while other groups showed their highest activity levels around later part of the day (between 10.00 am and 1.00 pm). After early morning, the active-high sedentary participants showed a sharp decline in their activity level during daytime hours (between 10.00 am and 3.00 pm), followed by a small peak in the evening (around 7.00 pm). On the other hand, the active-low sedentary participants sustained their peak activity levels for a longer duration (between 10.00 am and 1.00 pm) and showed a higher average VM cpm than all other groups over 24 h. When compared to the active-high sedentary participants, the highest peak in activity level was smaller in the inactive-high sedentary group (occurring around noon), but their overall activity level was higher throughout the day than their active-high sedentary counterparts. The distribution of the average PA levels, i.e., the daily average times accumulated in SB, LPA and MVPA (expressed as % of total wake hours), for all mutually exclusive categories is presented in Figure 3.

Table 2 summarizes the descriptive statistics and results from univariate analyses (i.e., parametric: one-way ANOVA; non-parametric: Kruskal–Wallis test) for the bioelectrical impedance analysis and fall risk assessment across four mutually exclusive groups. The body composition parameters are presented as absolute values and height-normalized indices (i.e., divided by the square of height). The mean (SD) body fat mass in the inactive-high sedentary group was 28.7 (13.9) kg, which was significantly higher than both the active-low sedentary (18.2 (12.6) kg, *p* = 0.010) and inactive-low sedentary participants (17.2 (3.66) kg, *p* = 0.027).

Figure 4 presents the scatterplot of the body fat mass index (BFMI) across the sedentary statuses (i.e., ratio of average SB time and average LPA time), stratified by active and inactive participants. As illustrated in Figure 4, the average BFMI in the inactive-high sedentary group was significantly higher compared to the other three PA groups (active-low sedentary: *p* = 0.002; active-high sedentary: *p* = 0.041; and inactive-low sedentary: *p* = 0.010). However, in the current sample, no significant group differences were observed for the extracellular-to-intracellular water ratio, extracellular-to-total body water ratio and lean mass index results.

In comparison to the inactive-high sedentary participants, the average skeletal muscle mass was significantly higher in the active-low sedentary (*p* = 0.010) and inactive-low sedentary (*p* = 0.032) groups, but not in the active-high sedentary category. Similarly, the average skeletal muscle mass index (SMI) was observed to be lower in the inactive-high sedentary participants than the active-low sedentary (*p* = 0.059) and inactive-low sedentary (*p* = 0.011) groups. Additionally, the average SMI was significantly higher in the inactive-low sedentary participants compared to their active-high sedentary counterparts (10.3 kg/m^2^ vs. 9.22 kg/m^2^, *p* = 0.023).

Compared to the inactive-low sedentary group, significantly lower appendicular lean mass index (ALMI) was observed in the active-high sedentary (*p* = 0.029) and inactive-high sedentary (*p* = 0.014) participants. Additionally, more favorable appendicular fat mass index (AFMI) values were found in the active-low sedentary (*p* = 0.007) and inactive-low sedentary (*p* = 0.031) groups than the inactive-high sedentary participants.

No significant group differences were found in the fear of falling and static balance scores in the current sample. However, the dynamic balance score (i.e., sit-to-stand performance) was significantly lower in the inactive-high sedentary participants than those categorized as active-low sedentary (*p* = 0.006) and inactive-low sedentary (*p* = 0.042). Figure 5 shows the scatterplot of the sit-to-stand scores across sedentary statuses (i.e., ratio of average SB time and average LPA time), grouped by active and inactive participants.

Table 3 presents the linear regression models for different outcome variables, adjusted for covariates (the unadjusted regression models are included in the Appendix A, Appendix A). The adjusted analyses showed that compared to the reference group (i.e., inactive-high sedentary group), the active-low sedentary participants had significantly lower body fat mass (*β* = −10.46, *p* = 0.006), BFMI (*β* = −4.37, *p* = 0.002), appendicular fat mass (*β* = −11.67, *p* = 0.008) and AFMI (*β* = −2.19, *p* = 0.003) scores, and higher skeletal muscle mass (*β* = 4.79, *p* = 0.004), SMI (*β* = 1.23, *p* = 0.017), appendicular lean mass (*β* = 6.38, *p* = 0.001) and ALMI (*β* = 1.89, *p* = 0.003) scores.

Additionally, more favorable outcomes in terms of body fat mass (*β* = −7.40, *p* = 0.019), BFMI (*β* = −3.14, *p* = 0.007), skeletal muscle mass (*β* = 3.64, *p* = 0.008), SMI (*β* = 1.05, *p* = 0.014), appendicular fat mass (*β* = −9.27, *p* = 0.011) and AFMI (*β* = −1.74, *p* = 0.005) were observed in the inactive-low sedentary group compared to the inactive-high sedentary category. The active-high sedentary participants only showed significantly higher AFMI (*β* = −1.69, *p* = 0.037) scores than the reference group. In addition, significantly higher dynamic balance scores were observed in the active-low sedentary (*β* = 4.52, *p* = 0.014) and inactive-low sedentary (*β* = 3.28, *p* = 0.031) groups in comparison to the reference group.

## 4. Discussion

This is the first study, to our knowledge, to evaluate the associations of mutually exclusive categories of wrist-worn accelerometer-based PA and SB with body composition and fall risk parameters in a sample of older US women. Overall, the less sedentary participants, irrespective of their MVPA status, showed more favorable BFMI, SMI and AFMI compared to those who were inactive and had high SB-LPA time ratios. Participants meeting MVPA recommendations, regardless of their sedentary status, had lower BFMI and AFMI scores in comparison to the inactive-high sedentary group. Additionally, the static balance scores were not significantly different across four mutually exclusive PA groups, but better dynamic balance ability was associated with the low sedentary status, even in those not meeting the MVPA guidelines.

Prior research has demonstrated the detrimental effects of SB on different health markers, including body composition and dynamic balance among older adults [12,13,15,16,17]. Additionally, there has been emerging evidence of the health-enhancing role of LPA in reducing mortality risk and maintaining favorable cardiometabolic biomarkers in older adults [45,46]. These previous analyses have been able to identify the individual effects of SB and LPA while adjusting for the time spent in MVPA. However, since an individual’s waking time is divided among SB, LPA and MVPA during any given day, it is important to understand the daily equilibrium between these constructs and their synergistic relationship with body composition and functional balance. Our findings indicate that, for participants meeting MVPA guidelines, no significant group differences were observed between active-low sedentary and active-high sedentary individuals for any outcome variable. However, for participants not meeting the MVPA recommendations, more favorable body composition profile and dynamic balance scores were observed in inactive-low sedentary participants compared to those classified as inactive-high sedentary. This is in broad agreement with the growing body of evidence that the MVPA status may influence the relationship between SB and health outcomes in such a way that SB only emerges as a determinant of health in individuals not meeting MVPA guidelines [20,21,47,48]. This suggests that meeting MVPA recommendations might provide some protective effects against the negative consequences of habitual SB among older adults.

Previous studies have reported lower MVPA to be linked with greater obesity levels in older adults [49,50]. Our results showed that for individuals with the high sedentary status (i.e., high SB-to-LPA time ratio), lower BFMI and AFMI scores were found in the active-high sedentary group in comparison to the inactive-high sedentary participants. However, for participants with the low sedentary status (i.e., low SB-to-LPA time ratio), no significant group differences were observed between the active-low sedentary and inactive-low sedentary groups for obesity markers (i.e., BFMI and AFMI). These findings suggest the importance of the daily balance between the SB and LPA times, which might buffer some of the adverse outcomes of insufficient MVPA in preserving healthy body composition. From a public health perspective, this knowledge can be relevant in managing and preventing obesity (in conjunction with healthy dietary behaviors) in older individuals who cannot meet MVPA guidelines due to chronic health conditions and low cardiorespiratory fitness, emphasizing the need to make LPA a pragmatic target in their PA interventions to combat sedentary lifestyles.

Interestingly, we observed that participants who were inactive and less sedentary had better SMI and ALMI scores compared to those that were active but highly sedentary in the current sample. This suggests that maintaining daily equilibrium of SB, LPA and MVPA durations among older adults might contribute to the attenuation of age-related losses in skeletal muscle mass. If confirmed in large-scale population-based cohort studies, these findings will have practical implications in the prevention of sarcopenia (i.e., the aging-associated decline in muscle mass and function) [51] among those older individuals who spend the majority of their waking hours in prolonged sitting. Therefore, it can be inferred from our results that sustainable, effective policy frameworks to counteract or reverse sarcopenia and maintain physical function in older adults should focus on developing guidelines on the recommended proportions of SB, LPA and MVPA on a given day (rather than only focusing on MVPA). Future prospective studies with large-scale nationally representative samples are required to further investigate such associations with different daily combinations of SB, LPA and MVPA to identify the optimal proportion of these constructs that can potentially aid in preserving muscle mass in older individuals.

We found that compared to inactive-high sedentary participants, significantly higher sit-to-stand scores, indicative of lower fall risks, were obtained in both active-low sedentary and inactive-high sedentary participants. Prior studies have reported greater time spent in SB to be negatively associated with dynamic balance among older adults [15,16], and our results are consistent with that. These results emphasize the need to develop multimodal PA interventions, combining balance exercises and strategies for reducing sedentary behavior, to promote fall prevention and functional mobility among older women. However, we did not observe any significant association between fear of falling and mutually exclusive PA categories in the current sample. The fear of falling prevalence in older adults can be influenced by different factors, such as their fall history, frailty or physical function [52]. Prior research has reported fear of falling being not independently associated with the total daily PA volume in community-dwelling older adults when accounting for physical function [53]. Therefore, future studies might consider examining the association between fear of falling and mutually exclusive PA categories, stratified by their fall history or functional status. 

Based on our study findings, it can be conferred that PA interventions to promote healthy body composition and dynamic balance in older adults should integrate approaches to combine both LPA and MVPA recommendations. Since LPA already accounts for a substantial portion of older adults’ daily activities [54], increasing LPA might provide a feasible target for enhancing the daily balance between sedentary time and total PA (i.e., LPA + MVPA) among older adults, especially for those with chronic diseases and geriatric conditions. Additionally, in our current sample, inactive-high sedentary participants had the least favorable outcomes in body composition profile. Therefore, gradual stepwise intervention strategies can be proposed to shift an older adult from the least desirable PA group (inactive-high sedentary) to the most desirable category (active-low sedentary). For instance, if a person is in the least desirable PA category, targeted interventions should first focus on replacing their sedentary time with LPA to move them to the inactive-low sedentary category to promote healthy lifestyle behaviors.

Our findings provide evidence to develop informed strategies for promoting the daily equilibrium between SB, LPA and MVPA among older adults to achieve more favorable body composition and dynamic balance results. However, our study has some limitations that should be mentioned. Firstly, the sedentary status was defined based on a conservative data-driven approach based on the distribution of sedentary time and LPA time scores in our study sample. This makes it difficult to define a specific threshold for the SB-to-LPA time ratio that can be applied to the general older adult population for targeted interventions. Secondly, our study investigations were only limited to the SB and PA domains, and the effect of the sleep duration was not considered. To date, few studies have reported the associations between sleep duration (longer or shorter) and body composition among adults [55,56]. Thus, future studies can consider exploring the combined effect of the total PA, SB, and sleep duration on body composition among older adults to develop recommendations on how their daily time should be allocated for LPA, MVPA and sleep. Thirdly, the cross-sectional nature of the study did not allow us to determine the casual relationships between variables. Fourthly, the sample size was relatively small and 75% of the participants were white. The small, non-representative nature of the sample limits the generalizability of our findings. Lastly, utilizing wrist-worn accelerometry devices might sometimes lead to the underestimation or overestimation of PA levels, since they cannot reliably detect non-ambulatory activities (such as cycling) or distinguish between different postures (sitting and standing).

## 5. Conclusions

In conclusion, the low sedentary status was associated with a more favorable body composition profile and reduced fall risk (i.e., dynamic balance) among our study participants, even in those not meeting the MVPA guidelines. Additionally, meeting the MVPA guidelines, irrespective of the sedentary status, was associated with a better AFMI score. Our results suggest that in the current sample, meeting MVPA guidelines did not attenuate unfavorable health outcomes if a person retained a highly sedentary lifestyle. Therefore, the joint prescription of sufficient MVPA and replacing sedentary time with LPA can aid in promoting a positive shift toward a healthy body composition and dynamic balance among older adults. Future prospective studies should focus on identifying the optimal ratio of SB, LPA and MVPA for health benefits to better inform public health policies for effective PA interventions.

## Figures and Tables

**Figure 1 ijerph-20-03595-f001:**
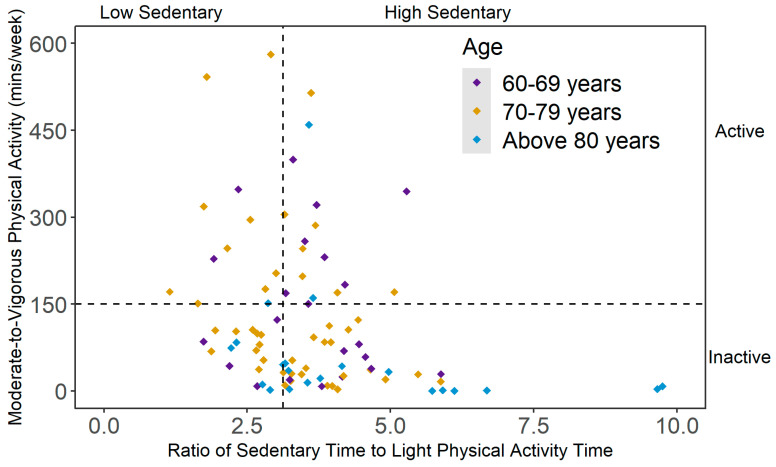
Scatterplot of mutually exclusive behavioral categories of physical activity (PA) and sedentary behavior (SB), stratified by age group. Note. Active = meeting 150 min/week of moderate-to-vigorous PA (MVPA); inactive = not meeting 150 min/week of MVPA; low sedentary = 1st tertile of the ratio between average sedentary time and average light PA (LPA) time; high sedentary = 2nd and 3rd tertiles of the ratio between average sedentary time and average LPA time.

**Figure 2 ijerph-20-03595-f002:**
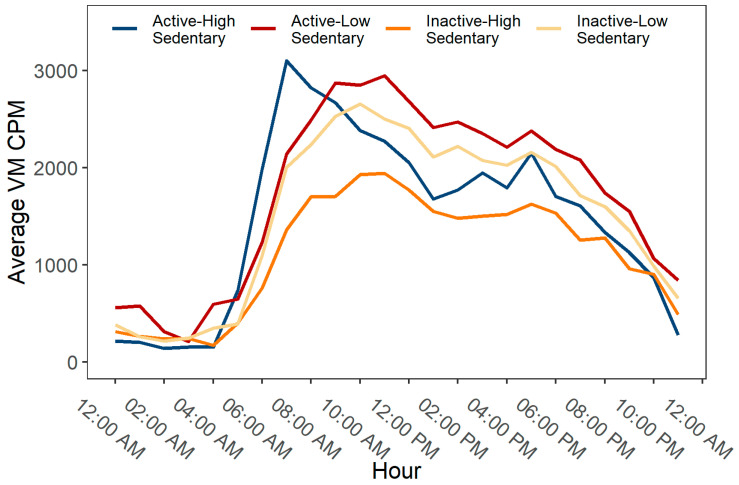
Hourly patterns of physical activity by mutually exclusive behavioral categories of physical activity (PA) and sedentary behavior (SB).

**Figure 3 ijerph-20-03595-f003:**
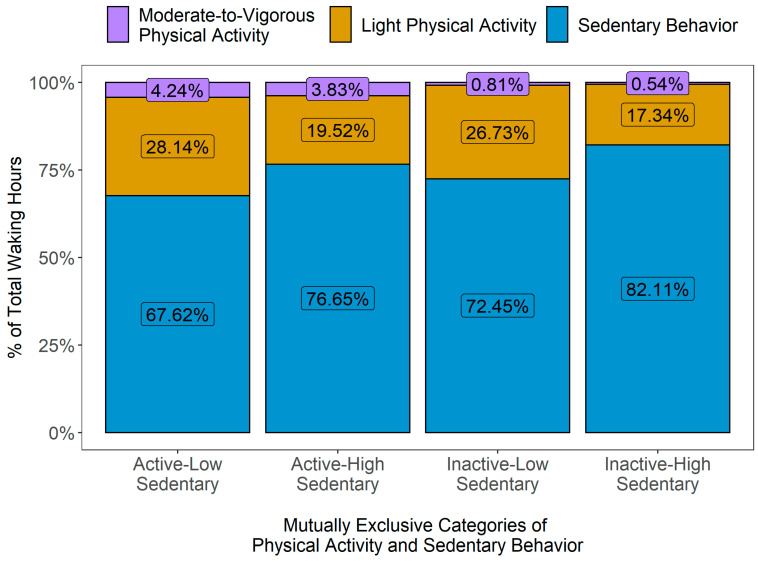
Distribution of average physical activity levels by mutually exclusive behavioral categories of physical activity (PA) and sedentary behavior (SB).

**Figure 4 ijerph-20-03595-f004:**
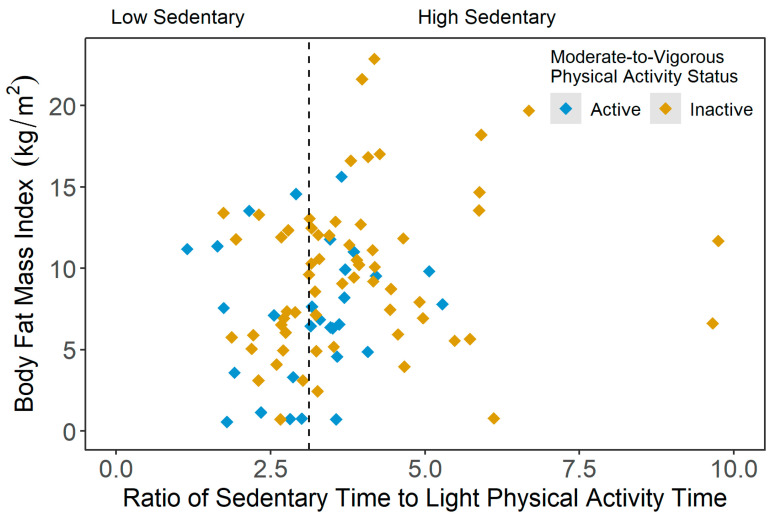
Scatterplot of body fat mass index (kg/m^2^) values across the ratio of average sedentary time and average light physical activity time results, stratified by moderate-to-vigorous physical activity (MVPA) status (active = meeting 150 min/week of moderate-to-vigorous PA (MVPA); inactive = not meeting 150 min/week of MVPA).

**Figure 5 ijerph-20-03595-f005:**
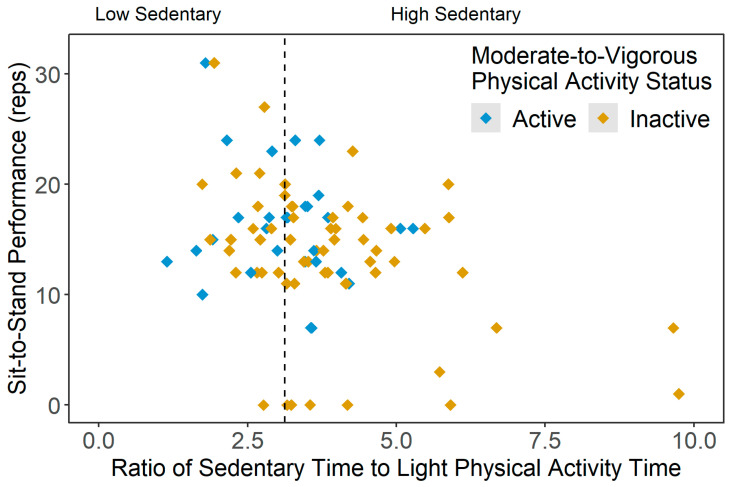
Scatterplot of dynamic balance scores (sit-to-stand scores, reps) across the ratio of average sedentary time and average light physical activity time results, stratified by moderate-to-vigorous physical activity (MVPA) status (active = meeting 150 min/week of moderate-to-vigorous PA (MVPA), inactive = not meeting 150 min/week of MVPA).

**Table 1 ijerph-20-03595-t001:** Characteristics of study participants, expressed as means (SD).

Variables	All	Active-Low Sedentary	Active-High Sedentary	Inactive-Low Sedentary	Inactive-High Sedentary	*p*
	*N* = 91	*n* = 17 (19%)	*n* = 12 (13%)	*n* = 29 (32%)	*n* = 33 (36%)	
Age (years)	74.9 (7.35)	73.4 (4.82)	71.8 (6.90)	76.2 (7.68)	75.7 (8.10)	0.259
Race, White, % (n) ^1^	75 (68)	88 (15)	75 (9)	76 (22)	67 (22)	0.424 ^2^
BMI (kg/m^2^)	26.8 (5.5)	24.5 (3.5)	25.1 (5.5)	26.8 (5.1)	28.6 (6.2)	**0.046**
SB time (min/day)	742 (130)	634 (87)	759 (71)	701 (103)	829 (131)	**<0.001**
LPA (min/day)	219 (63)	263 (61)	193 (28)	258 (47)	172 (45)	**<0.001**
MVPA (min/day)	17.4 (21.2)	42.1 (27.9)	38.1 (17.8)	7.9 (5.0)	5.6 (5.3)	**<0.001**

Note: ^1^ percentage (frequency); ^2^
*p* from chi-squared test. Bold indicates *p* < 0.05

**Table 2 ijerph-20-03595-t002:** Results from the bioelectrical impedance analysis and fall risk assessment.

Variables	All	Active-Low Sedentary	Active-High Sedentary	Inactive-Low Sedentary	Inactive-High Sedentary	*p*
Body composition						
Extracellular-to-intracellular water ratio (kg/kg)	0.631 (0.03)	0.630 (0.03)	0.632 (0.02)	0.623 (0.03)	0.636 (0.04)	0.630
Extracellular-to-total body water ratio (kg/kg)	0.386 (0.01)	0.386 (0.01)	0.387 (0.01)	0.384 (0.01)	0.387 (0.01)	0.593
Lean mass (kg)	46.9 (10.1)	46.0 (3.0)	47.5 (16.3)	47.5 (8.7)	45.7 (11.5)	0.802
Lean mass index (LMI, kg/m^2^)	17.3 (4.0)	16.7 (2.9)	17.5 (4.4)	17.2 (3.7)	17.4 (5.2)	0.966
Body fat mass (kg) ^1^	23.5 (12.8)	18.2 (12.6)	21.4 (10.1)	21.6 (10.8)	28.7 (13.9)	**0.021**
Body fat mass index (kg/m^2^) ^1^	8.9 (4.7)	6.7 (4.6)	7.9 (3.8)	8.0 (3.7)	11.1 (5.2)	**0.006**
Skeletal muscle mass (kg)	25.4 (6.0)	26.4 (6.6)	24.9 (2.6)	27.1 (9.1)	24.3 (5.2)	**0.041**
Skeletal muscle mass index (SMI, kg/m^2^)	9.6 (1.6)	9.9 (1.8)	9.2 (1.6)	10.3 (1.5)	9.0 (1.6)	**0.020**
Appendicular lean mass (kg)	19.6 (5.5)	22.2 (6.4)	19.4 (1.9)	20.9 (7.0)	18.4 (4.0)	0.057
Appendicular lean mass index (ALMI, kg/m^2^)	7.3 (1.7)	7.9 (1.8)	7.0 (1.2)	7.9 (1.3)	7.1 (1.3)	**0.028**
Appendicular fat mass (kg)	21.6 (18.5)	18.1 (21.4)	21.3 (10.5)	19.8 (19.4)	25.6 (17.4)	0.068
Appendicular fat mass index (AFMI, kg/m^2^)	3.9 (2.7)	3.1 (3.5)	3.7 (1.6)	3.5 (2.6)	4.4 (3.4)	**0.024**
Fall risk						
Perceived fall risk: Fear of falling score	9 (5)	8 (2)	9 (2.5)	9 (4)	10 (5)	0.421
Static balance score: Centre of pressure path length (cm)	26 (14)	22 (9)	24 (13)	25 (10)	30 (14)	0.111
Dynamic balance score: Sit-to-stand performance (reps)	15 (5)	17 (4)	15 (5)	15 (6)	13 (5)	**0.035**

Note: ^1^ expressed as means (SD); the rest of the variables are presented as medians (IQR). Bold indicates *p* < 0.05.

**Table 3 ijerph-20-03595-t003:** Associations with body composition and balance tests—results from the multiple linear regression analysis.

Variables	Active-Low Sedentary		Active-High Sedentary		Inactive-Low Sedentary		Inactive-High Sedentary
	*β* (*SE*)	*p*	*β* (*SE*)	*p*	*β* (*SE*)	*p*	
Extracellular-to-intracellular water ratio (%)	−0.01 (1.1)	0.994	−0.09 (1.3)	0.941	−1.65 (1.0)	0.088	Ref.
Extracellular-to-total body water ratio (%)	−0.01 (0.9)	0.990	−0.01 (1.1)	0.986	−0.69 (0.8)	0.091	Ref.
Lean mass (kg)	−0.84 (2.6)	0.747	0.79 (2.9)	0.786	0.32 (2.2)	0.883	Ref.
Lean mass index (LMI, kg/m^2^)	−1.41 (1.1)	0.190	−0.62 (1.2)	0.604	−0.34 (0.9)	0.705	Ref.
Body fat mass (kg)	**−10.46 (3.7)**	**0.006**	−6.64 (4.1)	0.112	**−7.40 (3.1)**	**0.019**	Ref.
Body fat mass index (kg/m^2^)	**−4.37 (1.3)**	**0.002**	−2.96 (1.5)	0.054	**−3.14 (1.1)**	**0.007**	Ref.
Skeletal muscle mass (kg)	**4.79 (1.6)**	**0.004**	1.00 (1.8)	0.579	**3.64 (1.3)**	**0.008**	Ref.
Skeletal muscle mass index (SMI, kg/m^2^)	**1.23 (0.5)**	**0.017**	−0.04 (0.6)	0.947	**1.05 (0.4)**	**0.014**	Ref.
Appendicular lean mass (kg)	**6.38 (1.9)**	**0.001**	0.83 (2.1)	0.697	2.24 (1.6)	0.162	Ref.
Appendicular lean mass index (ALMI, kg/m^2^)	**1.89 (0.6)**	**0.003**	−0.004 (0.7)	0.994	0.60 (0.5)	0.249	Ref.
Appendicular fat mass (kg)	**−11.67 (4.3)**	**0.008**	−8.69 (4.8)	0.072	**−9.27 (3.6)**	**0.011**	Ref.
Appendicular fat mass index (AFMI, kg/m^2^)	**−2.19 (0.7)**	**0.003**	**−1.69 (0.8)**	0.037	**−1.74 (0.6)**	**0.005**	Ref.
Perceived fall risk: Fear of falling score	−1.54 (1.19)	0.197	−1.13 (1.32)	0.397	−0.17 (0.99)	0.862	Ref.
Static balance score: Centre of pressure path length (cm)	−6.35 (4.4)	0.152	−4.67 (4.9)	0.341	−5.96 (3.7)	0.107	Ref.
Dynamic balance score: Sit-to-stand performance (reps)	**4.52 (1.8)**	**0.014**	1.51 (2.0)	0.452	**3.28 (1.5)**	**0.031**	Ref.

Note: *β* = standardized regression coefficient; SE = standard error. Associations were adjusted for age (years), race and accelerometer wear time (minutes/day). Bold indicates *p* < 0.05.

## Data Availability

The data presented in this study are available from the corresponding author upon reasonable request.

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
