# Peer review of "Associations of Mutually Exclusive Categories of Physical Activity and Sedentary Behavior with Body Composition and Fall Risk in Older Women: A Cross-Sectional Study"

_ijerph, 2023, doi:10.3390/ijerph20043595_

Round 1

Reviewer 1 Report

Dear authors:

First of all I congratulate you on the research carried out. 

Next, I would like to suggest some areas of improvement for your manuscript:

- First of all, the trial should be registered with your NCT number on www.clinicaltrial.gov.

- In addition, you should include a flowchart of the CONSORT trial to observe the sample of patients and whether or not there have been losses during the intervention.

- The introduction section is quite complete, I would only add the possible costs caused by falls in the elderly.

- In the methods section, they describe the inclusion and exclusion criteria, but in my opinion they should expand the inclusion criteria, since walking with canes can substantially modify the risk of falls, and on the other hand, they should add exclusion criteria, since they are essential to obtain a more homogeneous sample for their study. Therefore, this point is fundamental.

- The results section is quite complete, I would only point out that the figures have a very low image quality and you have to make an effort to visualise their content.

- The discussion section, in my opinion, is quite complete and discusses each variable in relation to other studies carried out, as well as quite up-to-date studies. They include their limitations and strengths of their study, and perhaps they could take out the strengths, as they are perfectly observed.

- That said, the relationship between increased physical activity and decreased risk of falls is a proven fact in the scientific community, and in my view, there are hardly any novel findings in their study. Perhaps they should improve the methods section and even add one more variable in order to have a stronger manuscript.

Best regards.

Author Response

Dear Reviewer

We would like to appreciate your thoughtful review of our manuscript, titled “Associations of Mutually Exclusive Categories of Physical Activity and Sedentary Behavior with Body Composition and Fall Risk in Older Women: A Cross-sectional Study”.

We have responded to all your comments and have revised the manuscript accordingly. Please find a point-by-point response to your comments in the attachment.

We believe these revisions have improved our paper and hope you find this revised submission acceptable for publication in the International Journal of Environmental Research and Public Health (IJERPH). Please let us know if there are further questions or clarifications.

Thank you for your consideration.

Sincerely,

Joon-Hyuk Park, PhD (He/him)

Assistant Professor, Department of Mechanical and Aerospace Engineering

Disability, Aging and Technology Cluster

University of Central Florida

Reviewer 2 Report

Dear Authors,

This manuscript is very well analyzed.

However, I regret not using the ActiGraph wGT3X-BT. As you know, ACSM uses ActiGraph wGT3X-BT as the standard accelerometer.

Please indicate in your manuscript how accurate the ActiGraph GT9X Link is.

Author Response

Dear Reviewer

We would like to appreciate your thoughtful review of our manuscript, titled “Associations of Mutually Exclusive Categories of Physical Activity and Sedentary Behavior with Body Composition and Fall Risk in Older Women: A Cross-sectional Study”.

We have responded to your comments and have revised the manuscript accordingly. Please find a point-by-point response to your comment in the attachment.

We believe these revisions have improved our paper and hope you find this revised submission acceptable for publication in the International Journal of Environmental Research and Public Health (IJERPH). Please let us know if there are further questions or clarifications.

Thank you for your consideration.

Sincerely,

Joon-Hyuk Park, PhD (He/him)

Assistant Professor, Department of Mechanical and Aerospace Engineering

Disability, Aging and Technology Cluster

University of Central Florida

Reviewer 3 Report

"All motion counts!" states a poster I have seen in a physiotherapy clinic. This manuscript confirms that claim after a study of 94 women aged 60-96, assessed for strenght and balance and other parameters, and their daily motion monitored with a wrist-borne accelerometer for 6 days. The results are presented through a number of tables and a couple of figures. 

However, the extensive data could also be presented in complementary ways that would be much more informative to scientifically literate patients and others.

For a start, I would be very interested in seeing "typical" activity graphs for "typical" participants for each of the four different boxes, to get a feeling for their different approaches to a day.

Second, I would like to see individual dots in a scatter plot, inserted the four-field graph in figure 1 (or separately if needed). The horizontal axis could then show the ratio between average sedentary time and average Light PA time, and the vertical axis could show minutes per week of MVPA. Possibly the dots in the scatter plot could also be shape and/or colour-coded for different age groups. (E.g. 60-64, 65-69, ... 90-96, alternatively 60-69, 70-79, 80-89, 90- ). (This plot could provide additional justification for the categorization used.)

Similarily, Figure 2 could show a scatterplot of BMFI and Figure 3 could show a scatterplot of individual sit/stand scores versus the Sedentary/LPA ratio, preferrably with different symbols for participants classified as active or inactive. 

I thus find the results interesting and worth publishing, but that these interesting results deserve a better visual presentation in addition to the tables and current figures.

Author Response

(The authors gave the same response as above.)

Round 2

Reviewer 1 Report

Dear authors:

First of all I would like to congratulate you on your research. Then I would like to suggest some areas for improvement:

-In the methods section, you should introduce some more pathology in the exclusion criteria, such as rheumatological diseases and lower limb dysmetria, as the registration will vary in this type of patients.

- They should add the criterion of validation of the devices used in the study. They should also register their study at www.clinicaltrials.gov where they will be assigned a study identification number (NCT....).

- The conclusions are not new.

- The rest of the sections comply with the regulations and are of sufficient quality to be published.

Best regards.

Author Response

Dear Reviewer

We would like to appreciate your thoughtful review of our manuscript, titled “Associations of Mutually Exclusive Categories of Physical Activity and Sedentary Behavior with Body Composition and Fall Risk in Older Women: A Cross-sectional Study”.

We have responded to all your comments and have revised the manuscript accordingly. Please find a point-by-point response to your comments in the attachment.

We believe these revisions have improved our paper and hope you find this revised submission acceptable for publication in the International Journal of Environmental Research and Public Health (IJERPH). Please let us know if there are further questions or clarifications.

Thank you for your consideration.

Sincerely,

Joon-Hyuk Park, PhD (He/Him)

Assistant Professor, Department of Mechanical Engineering

Disability, Aging and Technology Cluster

University of Central Florida

Reviewer 2 Report

Dear Authors,

Thank you for your reply. I fully understand the responses of the authors of this manuscript.

Author Response

Dear Reviewer

Thank you for your kind response. We appreciate your time and effort towards improving our manuscript.

Sincerely,
Joon-Hyuk Park, PhD (He/Him)
Assistant Professor, Department of Mechanical and Aerospace Engineering
Disability, Aging and Technology Cluster
University of Central Florida